# CRISPR Gene Therapy: A Promising One-Time Therapeutic Approach for Transfusion-Dependent β-Thalassemia— CRISPR-Cas9 Gene Editing for β-Thalassemia

**Udani Gamage [1,2], Kesari Warnakulasuriya [1,3], Sonali Hansika [1] and Gayathri N. Silva [1,*]**

1 Department of Chemistry, University of Colombo, Colombo 00700, Sri Lanka
2 Molecular and Cellular Biology Program, Department of Chemistry and Biochemistry, Ohio University, Athens, OH 45701, USA
3 School of Molecular Sciences, Arizona State University, Tempe, AZ 85281, USA
* Correspondence: gayathris@chem.cmb.ac.lk; Fax: +94-11-250-3367

**Abstract:** β-Thalassemia is an inherited hematological disorder that results from genetic changes in the β-globin gene, leading to the reduced or absent synthesis of β-globin. For several decades, the only curative treatment option for β-thalassemia has been allogeneic hematopoietic cell transplantation (allo-HCT). Nonetheless, rapid progress in genome modification technologies holds great potential for treating this disease and will soon change the current standard of care for β-thalassemia. For instance, the emergence of the CRISPR/Cas9 genome editing platform has opened the door for precision gene editing and can serve as an effective molecular treatment for a multitude of genetic diseases. Investigational studies were carried out to treat β-thalassemia patients utilizing CRISPR-based CTX001 therapy targeting the fetal hemoglobin silencer BCL11A to restore γ-globin expression in place of deficient β-globin. The results of recently carried out clinical trials provide hope of CTX001 being a promising one-time therapeutic option to treat β-hemoglobinopathies. This review provides an insight into the key scientific steps that led to the development and application of novel CRISPR/Cas9–based gene therapies as a promising therapeutic platform for transfusion-dependent β-thalassemia (TDT). Despite the resulting ethical, moral, and social challenges, CRISPR provides an excellent treatment option against hemoglobin-associated genetic diseases.

**Keywords:** β-thalassemia; β-globin; γ-globin; BCL11A; CRISPR/Cas9 technology; CTX001 therapy

## 1. Introduction

Thalassemia is a hereditary hematological disorder that has a typical autosomal recessive pattern of inheritance [1]. Approximately 4.4 of every 10,000 live births are affected by the α- or β-thalassemia trait, which account for 1.7% of the world's population [2,3]. This syndrome is identified as a life-threatening disease in developing countries where the lack of early diagnosis and genetic counseling have contributed to the widespread maintenance of the disease in the population [4,5]. α-Thalassemia is prevalent in African and Southeast Asian populations, whereas β-thalassemia is prevalent in Mediterranean countries, Southeast Asia, Middle East, African countries, and South America [6,7]. The epidemiology of thalassemia is continuously changing with the migration patterns and interracial marriages, causing the disease to spread into the regions where it was previously absent. The widespread transmission of the disease has now become a global health concern and requires innovative new treatments that can help slow its spread [4].

β-thalassemia is classified as thalassemia major, minor, and intermedia based on the zygosity of the β-globin gene (HBB) mutations. The homozygous mutation ($β^0$) is characterized by the total absence of β-globin chains, causing β-thalassemia major or Cooley's anemia [8]. Cooley's anemia is the most severe form of β-thalassemia. It is characterized by the inheritance of two copies of the defective β-globin gene, which is

essential to produce the severe clinical features of the disease [9]. Children who inherit homozygous mutations are healthy at birth but usually develop major symptoms of the disease after only 6 months of life. At this stage of development, changes in transcriptional regulation and chromatin remodeling cause fetal-to-adult hemoglobin switch, leading to silencing of the fetal globin genes [8,10]. Thus, clinical manifestations of the homozygous mutations in HBB only become evident on completion of the switching process [11]. In patients with severe disease, the absence of β-globin leads to the accumulation of unpaired α-globin chains. The excess α-globin aggregates to form precipitates in erythroid precursor cells, damaging the cell membrane and facilitating apoptosis of the erythroid precursor cells (ineffective erythropoiesis) [12]. Thus, individuals with β-thalassemia major have severe anemia and are required to undergo lifelong blood transfusions.

A heterozygous mutation (β+) causes β-thalassemia minor, resulting in an asymptomatic individual with mild to moderate microcytic anemia. Patients whose clinical severity lies between those of thalassemia major and thalassemia minor are categorized as having thalassemia intermedia [13]. Thalassemia intermedia is genotypically heterogenous [11]. Most of the thalassemia intermedia patients are homozygotes or compound heterozygous for β-thalassemia [9]. In some instances, only a single β-globin locus is affected while other loci are completely normal, resulting in a dominantly inherited thalassemia intermedia [14,15]. Clinically, β-thalassemia intermedia patients show mild anemia and do not require lifelong transfusions or chelation therapy [16,17]. Occasionally, thalassemia intermedia patients undergo blood transfusions but less frequently compared to regularly-transfused individuals with β-thalassemia major [11,16].

Recent data estimated that ~600,000 symptomatic children are born with β-thalassemia annually, and approximately 80–90 million individuals of the global population are carriers of the disease [9]. Individuals with β-thalassemia experience serious clinical, economical, psychological, and social challenges throughout their life as the nature of the standard treatments available for severe and moderate forms of thalassemia impose a heavy economic and psychosocial burden on these patients and their families [16,18]. The standard treatment options that are widely used for the management of β-thalassemia include blood transfusion and iron chelation therapy [19,20]. Although their clinical applications are widespread, these methods have significant limitations and challenges. For instance, repeated blood transfusions can lead to iron overload with life-threatening complications, such as heart failure, endocrine dysfunction, and liver disease. In contrast, lack of adequate blood transfusion results in the reduction of life expectancy of transfusion-dependent thalassemia (TDT) patients. Thus, there is a compelling necessity for the development and screening of novel therapeutic approaches that hold promise as a one-time therapeutic option for the complete cure of the disease, improving the survival and quality of life of β-thalassemia patients [19,21].

Recent advances in understanding the pathophysiology of β-thalassemia have facilitated the development of new therapeutic strategies for the disease. Novel approaches were developed to correct the α- and β-globin chain imbalance to resolve ineffective erythropoiesis and chronic iron overload [19]. For example, pharmacological compounds such as hydroxyurea, which acts on several signaling pathways to decrease DNA methylation at the promoter regions of γ-globin genes to augment γ-globin expression, are being recommended to treat β-thalassemia. An increase in γ-globin levels facilitates fetal hemoglobin (HbF) production and ameliorates the severity of the disease, reducing the requirement for frequent blood transfusions [22–24]. Moreover, hydroxyurea increases the total hemoglobin levels in patients and is proven to be a suitable non-curative treatment strategy for β-thalassemia. Hydroxyurea is found to be effective in treating thalassemia intermedia and its effectiveness in treating thalassemia major is yet to be confirmed via randomized clinical trials [25–27]. In addition, recombinant fusion protein Luspatercept (ACE-536), Sotatercept (ACE-011), and compounds that inhibit hepcidin and ferroportin are some of the novel pharmacological strategies that target ineffective erythropoiesis to ameliorate the clinical manifestations of the disease [28–30].

Allogeneic hematopoietic stem cell transplantation (HSCT) was the only effective and potentially curative therapy available for β-thalassemia major for several decades. In HSCT, allogenic stem cells are used as vectors to correct the genetic defects in β-thalassemia by introducing the essential wild-type HBB for normal hematopoiesis [31]. However, access to this therapy is limited due to the lack of human leukocyte antigen (HLA)-matched donors. Patients may also develop long-term serious complications including immune-mediated diseases, endocrine disorders, and impaired pulmonary and respiratory functions after receiving HSCT therapy [32–35]. Splenectomy is another treatment modality that is performed to decrease the transfusion requirement in patients with thalassemia major. However, it is associated with unfavorable consequences such as cardiac failure, retardation of growth and sexual development, and increased susceptibility to infections [36].

Over the last few decades, much effort has been put into developing one-time curative therapeutic approaches for β-thalassemia where the patient does not require any further treatment after a single treatment is completed. Gene therapy approaches for β-thalassemia have gained significant attention as a realistic and effective therapeutic approach to achieve sustainable, stable, and high-level expression of functional globin genes in TDT patients with no mortality and severe immunological complications such as graft rejection and clonal dominance [9,34,37,38].

## 2. Gene Therapy as a Promising Cure for the Acute form of β-Thalassemia

Current gene therapy approaches offer great therapeutic promise with the most remarkable clinical outcomes for inherited or acquired disorders including cancer, viral infections, and recessive genetic disorders such as thalassemia, sickle cell anemia, cystic fibrosis and hemophilia [39,40]. The goal of this technique is to correct defective genes by administering functional genetic material into cells to produce a lasting therapeutic effect. Gene therapy strategies can be divided into two main categories based on their mode of gene delivery, namely, ex vivo and in vivo gene delivery. Ex vivo gene therapy is a novel approach that entails harvesting cells from the patient, genetically modifying them in a laboratory and transplanting the tailored cells back into the target tissue. This method offers a major advantage over the in vivo gene delivery method due to the ability to fully characterize and eliminate deleterious properties of the altered cells before transplantation [41]. The transplanted genetically modified cells function in secreting and disseminating targeted proteins into the surrounding environment [42]. In contrast, the in vivo strategy delivers DNA directly to resident cells of the target tissue, typically via a viral vector [40]. Replication-deficient viral vectors (e.g., adenoviruses, adenoviral associated viruses, retroviruses, and lentiviruses) can function as gene delivery vehicles to introduce genetic material to target cells such as germline and somatic cells [43]. This technology is now being widely applied in clinical trials to treat both inherited and acquired disorders [42].

Viral vectors serve as a popular gene delivery method for β-thalassemia. For instance, lentiviruses were engineered to be excellent vector candidates for the efficient transfer of HBB in both human and animal models. In multiple mouse and primate models with β-thalassemia major [44] and intermedia, lentiviral mediated β-globin gene transfer leads to alleviation of anemia and subsequent organ damage, making recombinant lentiviruses one of the most effective vector systems for β-thalassemia gene therapy [45,46]. Lentiviral vectors are considered a successful application for clinical trials as a result of their ability for efficient transduction of cells, such as CD34+, that undergo limited proliferation, as well as non-dividing cells with enhanced genomic stability [46,47]. In 2006, human clinical research was launched (LG001 study) utilizing lentiviral vectors for β-hemoglobinopathies [47,48]. HPV569 lentiviral vector 35 was employed to transfer the β-globin gene comprising the β-globin locus control region (LCR), which regulates the expression of the β-globin gene in autologous hematopoietic stem cells of β-thalassemic patients [48]. This therapy was clinically successful and eliminated the requirement for long-term transfusion after 1 year and was sustained for almost 8 years. The total hemoglobin level was maintained at a steady

state around 8 g/dL after 2–8 years of the treatment. Furthermore, non-hematological or drug-product-related adverse events were not reported upon infusion [49].

A recent comprehensive study ensures the viability of lentiglobin BB305 gene therapy, which transduces the βA-T87Q-globin gene that codes for T87Q amino acid substituted HbA into autologous CD34+ HSPC. Ex vivo modified cells are then infused back into β-thalassemia major patients to facilitate sustained expression of the engineered β-globin [48,50]. Betibeglogene autotemcel (ZYNTEGLO™), developed by Bluebird Bio, is classified as an advanced therapeutic medicinal product (ATMP) and is considered to be the first approved gene therapy manufactured using lentiglobin BB305 for the treatment of TDT [51]. According to the Pharmacovigilance Risk Assessment Committee (PRAC) report put forwarded by the European Medicines Agency (EMA), ZYNTEGLO™ was granted conditional approval by the European Union (EU) in June 2019 for the clinical management of β-thalassemia patients of 12 years and older suffering with TDT or for those who contain a non-β0/β0 genotype [52,53]. Clinical investigations from the North Star study at phases HGB-204, HGB-205, HGB-207, and HGB-212 employed 63 patients (HGB 207 and HGB 212 are still in progress) while, presently, follow-up safety protocols are evaluated to ensure long-term stability and integrity of the therapy (LTF-303 study) [54,55]. ZYNTEGLO™ is considered a one-time genetic treatment that introduces functional, engineered copies of β-globin genes (βA-T87Q) through transduction of autologous hematopoietic stem cells with a replication-incompetent, self-inactivating lentiviral vector BB305 [56]. The dosage of ZYNTEGLO™ is determined according to the body weight of an individual, and the administration is predominantly by intravenous injection. Preparatory clinical studies demonstrated that the modified β-globin was produced at significant levels to maintain the normal level of adult hemoglobin in the patients, possibly eliminating the need for red cell transfusions. The European Commission (EC) has designated this medication for the treatment of intermediate and severe β-thalassemia as an orphan pharmaceutical product. The US Food and Drug Administration (FDA) has also granted orphan drug status for ZYNTEGLO™ for the treatment of TDT [53].

Although lentiviral vectors belong to an RNA-based viral vector platform, they can be modified to diminish their pathogenicity. Clinical investigations are currently undertaken to assess the possible application of third-generation lentiviral vectors and engineering them to effectively transfer genetic material to target cells while continuing long-term stable expression [57]. Engineering self-inactivating lentiviral vectors manifest the removal of viral long terminal repeats, leading to enhanced cargo capacity, which can accommodate the altered target gene and its regulatory elements to augment the transgene expression [44]. Furthermore, non-clinical studies in mouse models have revealed minor concerns of genotoxic effects due to the trans-activation of genetic elements following the transfer of lentiviral vectors carrying the human β-globin gene [58]. However, according to the guidelines enforced by regulatory authorities such as FDA and EMA, the crucial scrutinization of certain parameters, such as vector toxicity and stable integration, is imperative for the safety and efficacy of lentiglobin gene therapy [59]. In February 2021, concerns came to light after evaluating a patient who received lentiglobin gene therapy and was diagnosed with acute myeloid leukemia (AML) [60]. Although the available data do not evoke many indications of insertional mutagenesis and tumorigenic risk of ZYNTEGLO™ gene therapy, PRAC recommends frequent (at least once a year) monitoring of patients for adverse conditions such as myelodysplasia, leukemia, or lymphoma [55].

While viral gene delivery methods can achieve a higher transfection efficiency, their application is limited due to safety and toxicity issues. For example, vector immunogenicity, the possibility of oncogenic transformation and sudden death of recipients have led to the replacement of viral-vector-based treatment with alternative gene therapy strategies [61,62]. Emergence of the Clustered Regularly Interspaced Short Palindromic Repeats/CRISPR-associated protein 9 (CRISPR/Cas9) genome engendering tool offers a feasible and elegant option for site-specific gene editing to introduce desired gene modifications in a safe and efficacious manner, alleviating some of the difficulties associated with traditional gene

therapy [63]. Recently, preliminary results of a CRISPR-phase-1 clinical trial have uplifted the hope for a possible breakthrough one-time curative option for β-thalassemia [64]. This review attempts to highlight the outstanding promise of the novel CRISPR-based gene therapy to treat TDT and the scientific approach taken by this method to treat hereditary hemoglobinopathies.

## 3. Advances in CRISPR Gene Therapy Hold Great Promise as an Effective One-Time Treatment Option for TDT

CRISPR is a breakthrough RNA-guided genome editing technology of the 21st century that holds a great therapeutic promise against hereditary blood diseases [21]. CRISPR-mediated adaptive immune systems were first discovered in bacteria that provide protection against invading foreign genetic elements [65–68]. The bacterial genome contains short repetitive sequences known as CRISPR arrays, separated by non-repetitive protospacer sequences that will be transcribed and processed into matured CRISPR RNA (crRNA) sequences that are homologous to the invading genetic material. Assembly of an active CRISPR/Cas effector complex requires the incorporation of mature crRNA that guides the effector complex to the invading nucleic acids and a separately encoded multi-domain nuclease called Cas (Class I nuclease) or a set of Cas nucleases (Class II nucleases) [67]. Target recognition is achieved by the extensive complementarity between the invader DNA and crRNA sequence, leading to the Cas-dependent double-stranded cleavage of the crRNA-foreign DNA complex [65,69]. The type II CRISPR system comprising a single effector nuclease Cas9 is the simplest and best understood CRISPR system that is widely adopted in genome engineering applications [65,66,69]. The Type II system requires a second scaffold RNA molecule called trans-activating crRNA (tracrRNA) that base pairs with crRNA to form a dual guide RNA structure (gRNA), which is essential for target recognition and crRNA-guided DNA cleavage, preceding the protospacer adjacent motif (PAM) present in the target sequence [67,70,71]. The remarkable programmable capacity of type II systems was harnessed to develop powerful gene editing tools to target any desired DNA sequence by customizing the crRNA sequence. While crRNA and tracrRNA are separately encoded in nature, scientists have engineered a chimeric single-guide RNA (sgRNA) molecule by fusing crRNA with tracrRNA. In CRISPR-based therapeutic approaches, safe and efficient delivery of the CRISPR/Cas9 system into target cells is an important consideration. CRISPR/Cas9 system can be introduced into isolated target cells from β-thalassemia patients through viral or non-viral delivery vectors where the RNA-dependent DNA cleavage is carried out by Cas9 after recognition of the target sequence by sgRNA. Electroporation and microinjection can be indicated as safe non-vector-based methods for CRISPR delivery [39,72,73].

A deep understanding of the pathophysiology of β-thalassemia led to the successful application of the CRISPR/Cas9 genome editing tool from bench to clinical trials for use as a promising curative therapeutic intervention for β-thalassemia. One of the widely studied gene therapy approaches for this disease is to restore the normal β-globin expression by correcting the disease-causing HBB mutations via CRISPR/Cas9 mediated homology-directed repair (HDR) [11,21]. Nonetheless, this approach is challenged by the presence of a wide range of mutations in HBB, as specific sgRNA and donor templates for each HBB mutation must be designed, optimized, and validated to be used in the therapy. However, scientists have successfully applied the CRISPR/Cas9 tool to correct HBB mutations in patient-derived induced pluripotent stem cells (iPSCs) [74]. A study aimed to correct two different HBB mutations (codon 41/42 with 4 bp deletion (-TCTT) and −28 with (A > G) substitution in the promoter) utilizing CRISPR/Cas9–mediated HDR resulted in the conversion of homozygous β-thalassemia to the heterozygous state and restored normal HBB expression in erythrocytes differentiated from the corrected iPSCs [75]. Genetic corrections in HBB using the same approach were observed in several other studies [75–79]. For instance, the CRISPR/Cas9 tool and the ssODN (single-strand oligodeoxynucleotide) donor template were utilized to genetically correct iPSCs retrieved from a patient who is

double heterozygous for hemoglobin E/β-thalassemia. The corrected clones were then differentiated into erythroid cells with restored expression of mature β-globin [77]. In a similar study, the CRISPR/Cas9 tool was utilized to correct the β-thalassemia iPSCs with CD17 (A > T) homozygous mutation in the HBB locus. Correction of the mutation resulted in the restoration of the normal β-globin expression with improved hematopoietic differentiation ability and relieving ineffective erythropoiesis [78,80]. Moreover, an in vivo study has demonstrated successful correction of HBB mutations (homozygous 41/42 deletion) in patient-specific iPSCs by restoring the β-globin expression in differentiated HSCs with no tumorigenesis, supporting the safe clinical application of CRISPR/Cas9 gene therapy for β-thalassemia [81].

Novel treatment strategies for TDT give significant attention to molecular mechanisms of globulin switching to permanently restore the α- and β-globin chain balance in patients [82,83]. CRISPR technology was successfully implemented to reactivate the previously silenced γ-globin gene to increase HbF production in TDT patients [7,82]. During the fetal stage, the synthesis of γ-globin balances the α-globin levels by combining to form adequate levels of HbF ($\alpha2\gamma2$), while the synthesis of β-globin replaces γ-globin to form adult hemoglobin (HbA, $\alpha2\beta2$) after birth (during the first year of life), which predominates in adult life [83]. In adult reticulocytes, the expression of the γ-globin gene is silenced by the heme-regulated inhibitor (HRI, also known as EIF2AK). HRI is an erythroid-specific kinase that can phosphorylate translation initiation factor eIF2α. Phosphorylation of eIF2α inhibits translation of γ-globin mRNA, with a concomitant decrease of HbF levels in adults [21,84]. The zinc finger transcription factor BCL11A (B cell lymphoma/leukemia 11A) is considered a critical mediator in silencing γ-globin expression and lowering HbF levels in adult human erythroid cells during γ- to β-globin switching. BCL11A works as a transcriptional repressor of γ-globin expression via direct binding of BCL11A to the TGACCA motif in the γ-globin promoter, silencing HbF production during fetal to adult hemoglobin switching [85–87] (Figure 1a).

The reduced BCL11A levels are found to decrease HRI production, ultimately enhancing the production of HbF [84]. Furthermore, naturally occurring mutations in the TGACCA motif are linked to the high persistence of HbF. Built on this evidence, scientists postulated that the alterations in the TGACCA motif would be sufficient to reactivate HbF production at adequate levels to ameliorate the severity of β-thalassemia [88]. Ex vivo and in vivo hematopoietic stem cell editing were performed using transgenic mice carrying the human β-globin locus (β-YAC) to inhibit BCL11A binding to the TGACCA motif in the γ-globin promoter, to increase the expression of γ-globin [87]. In this study, a helper-dependent human CD46-targeting adenovirus vector that comprises CRISPR/Cas9 components for the disruption of the BCL11A binding region of the γ-globin promoter was utilized. This study demonstrated an efficient cleavage of the target locus, resulting in a striking transition from β- to γ-globin expression without any hematological abnormalities [88].

Another mechanism adopted by BCL11A to repress the γ-globin expression involves binding to a γ-globin promoter via long-range interactions with multi-protein complexes (Figure 1b). BCL11A containing multi-subunit complexes typically contain different transcriptional co-repressors and chromatin-modifying subunits (e.g., Lysine-specific demethylase 1 and repressor element-1 silencing transcription factor corepressor 1 complex (LSD1/CoREST demethylase complex), DNA methyltransferase 1 (DNMT1) and nucleosome remodeling, deacetylase complex (NuRD), and SRY-Box transcription factor 6 (SOX6)). For instance, SOX6 represses γ-globin gene expression by binding to the proximal γ-globin promoters to facilitate the recruitment of BCL11A to its binding site via long-range protein–protein interactions [91,93]. The introduction of CRISPR/Cas9-mediated indel mutations to the SOX6 consensus within the γ-globin promoter prevented SOX6-γ-globin promoter interactions, consequently reactivating the γ-globin gene expression. CRISPR-mediated incorporation of three different indels in three different loci in the SOX6 consensus within the γ-globin promoter prevented SOX6-γ-globin promoter interactions in K562 cells. These

CRISPR-modified cells with indel percentages of 30%, 25%, and 24% have shown 1.3, 2.1, and 1.1 fold increments of γ-globin gene expression, respectively [94].

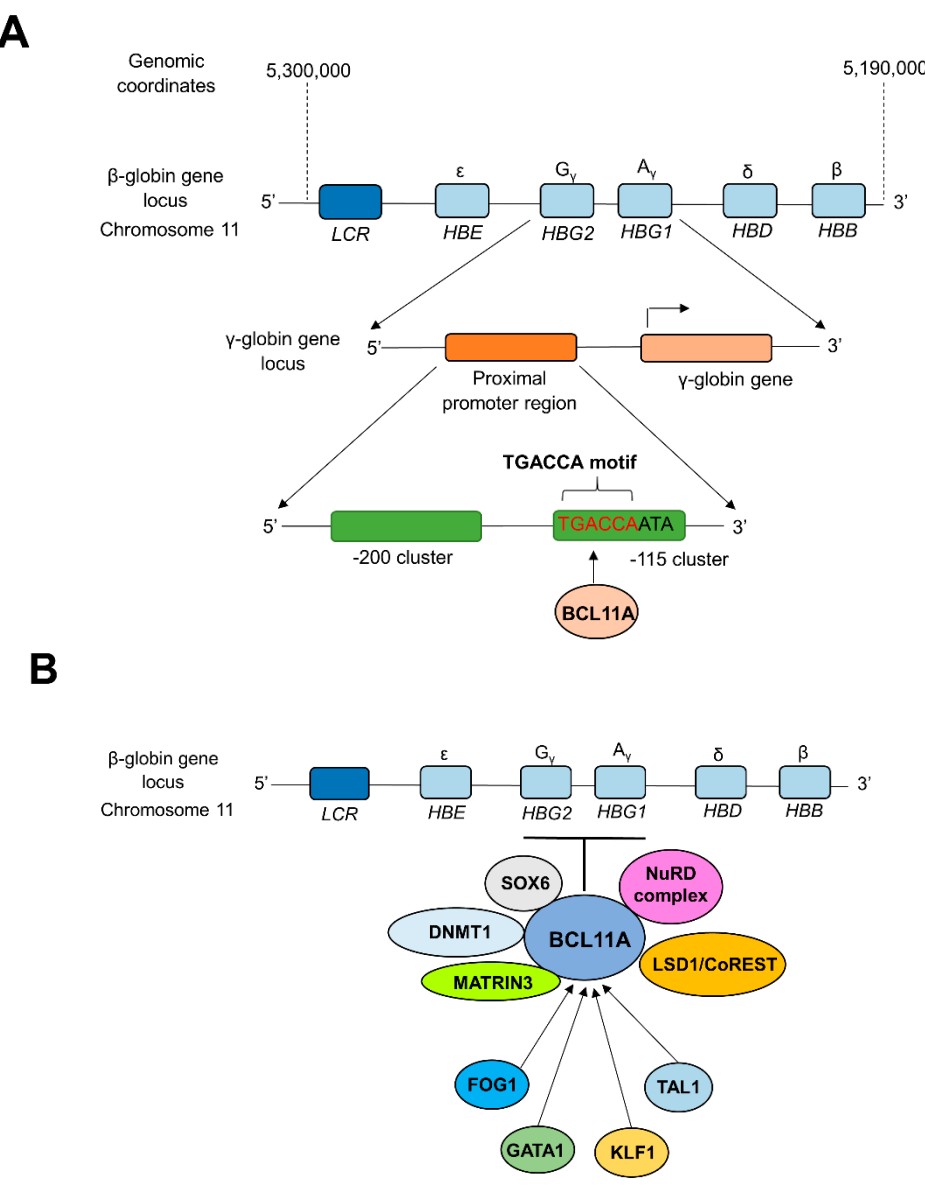

**Figure 1.** BCL11A-mediated transcriptional silencing of γ-globin. (**A**) Direct repression of the promoter region of γ-globin genes by BCL11A. Interactions between zinc finger motifs of BCL11A and the proximal TGACCA motif (−118 to −113) of γ-globin promoter silence the expression of γ-globin during fetal to adult hemoglobin switching [86,89]. (**B**) Long-range interactions of BCL11A with multi-protein complexes and transcription factors. The level of expression of BCL11A is controlled by several transcription factors such as FOG1, GATA1, TAL1, etc. At the same time, BCL11A is capable of repressing the γ-globin expression through long-range interactions and cooperation with multi-protein complexes such as NuRD, DNMT1, MATRIN3, etc. [90–92]. *LCR*; β-globin locus control region, *HBE*; Hemoglobin Subunit Epsilon gene, *HBG1*; Hemoglobin subunit Gamma 1 gene, *HBG2*; Hemoglobin Subunit Gamma 2 gene, *HBD*; Hemoglobin Subunit Delta gene, *HBA*; Hemoglobin Subunit Alpha gene. NuRD; Nucleosome Remodeling and Deacetylase, LSD1/CoREST; Lysine Specific Demethylase 1/Repressor Element 1 Silencing Transcription Factor Corepressor 1, MATRIN3; Nuclear Matrix Protein-3, DNMT1; DNA Methyltransferase 1, SOX6; SRY-box transcription factor 6, TAL1; T-cell Acute Lymphoblastic Leukemia 1, KLF1; Kruppel-Like Factor 1, GATA1; GATA binding protein 1, FOG1; Zinc finger protein FOG family member 1.

BCL11A expression is transcriptionally upregulated by binding transcriptional activators to the erythroid-specific enhancer region of BCL11A. GATA-1 is a zinc finger transcriptional activator that specifically binds to the GATA-1 transcriptional activator binding sequence (GATA motif) in the erythroid specific enhancer region of BCL11A to enhance its expression [95] (Figure 2).

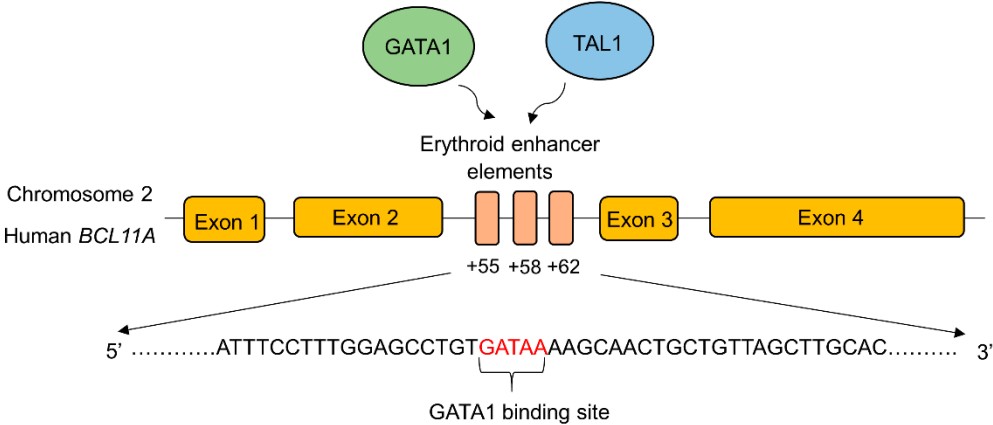

**Figure 2.** Schematic depiction of the transcription factor binding sites in the erythroid enhancer region of *BCL11A*. The binding of transcription factors (GATA1, TAL1) increases the *BCL11A* expression resulting in decreased γ-globin gene expression. Disruption of the erythroid-specific enhancer of *BCL11A* increased the endogenous expression of γ-globin and HbF production in adults. This approach is identified as an effective therapeutic strategy to treat β-hemoglobin disorders [96]. GATA1; GATA binding protein 1, TAL1; T-cell acute lymphoblastic leukemia 1. The major transcriptional activators of *BCL11A*, GATA1, and TAL1 bind to the erythroid enhancer elements in intron 2.

GATA-1 also works cooperatively with BCL11A as components of the NuRD complex during γ- to β-globin switching [97]. Therefore, the disruption of erythroid-specific enhancer region of BCL11A to reduce its expression is considered to be an effective therapeutic approach to ameliorating β-hemoglobinopathies. CRISPR/Cas9-mediated deletion of 200 bp in the erythroid-specific enhancer region [96] of BCL11A resulted in the downregulation of BCL11A expression and reactivation of γ-globin gene in K562 cells [98]. In another study, electroporation of Cas9 and 20 nt long sgRNA targeting the GATA-1 binding site of the +58 erythroid enhancer of BCL11A in CD34+ HSPCs generated a highly penetrant disruption of this motif, leading to a reduction in BCL11A transcript expression by 54.6%, and concomitant enhancement in γ-globin expression, ranging from 35.3–75.1%, in comparison to 14.7% in the unmodified cells [99,100]. Recently, similar work has been carried out to create indels in this motif using zinc finger nucleases (ZFN). The investigational cell therapy ST-400 comprised of an ex vivo, ZFN-mediated disruption of the GATA binding site of BCL11A in autologous CD34+ cells is currently being carried out in a phase 1/2a clinical trial to enhance HbF levels in CD34+ cells from β-thalassemic patients and human adult-stage erythroid cells. Safety and efficacy studies are expected to be followed for 3 years after infusion. Serious adverse events related to hypersensitivity were reported but were successfully managed under medical intervention [101].

The promoter region of BCL11A or its enhancer-binding site can serve as excellent targets to inhibit or downregulate BCL11A expression. Several studies have demonstrated the excellent promise of targeted disruption of the BCL11A erythroid-specific enhancer region by CRISPR/Cas9 system leading to a significant induction of γ-globin levels [99,100,102]. Emulating the productivity of this strategy, industry-sponsored clinical trials have paved the way to deliver advanced medical care for β-hemoglobinopathies.

## 4. CTX001 Therapy

CTX001 is an investigational genetically modified cell therapy studied by CRISPR therapeutics (Cambridge, MA, USA) and Vertex Pharmaceuticals (Boston, MA, USA) for inherited hematological disorders such as sickle cell disease (SCD) and TDT. Currently, CTX001 clinical trials are recruiting patients from the United States. It is a non-viral based, ex vivo CRISPR/Cas9-mediated site-specific editing in the erythroid-specific enhancer binding site of BCL11A in autologous CD34+ hematopoietic stem and progenitor cells (HSPC) to increase γ-globin gene expression and, subsequently, increase the HbF level in patients. The CTX001 infusion was shown to yield 80% allele modification in the BCL11A locus without evidence of off-target editing [64]. The upregulation of HbF reduced the transfusion requirement and anemic condition in β-thalassemia and lessened clinical complications in SCD patients, such as vaso-occlusive crises (VOCs) [103].

The first in-human clinical studies of CTX001 for treating SCD and TDT are CLIMB SCD-121 (NCT03745287) and CLIMB THAL-111 (NCT03655678), respectively [103]. CLIMB SCD-121 is a phase 1/2, multi-site, open-label, single-dose trial that investigates the safety and efficacy of the CRISPR/Cas9 modified autologous CD34+ HSPCs in SCD patients. Patients aged 12 to 35 years with severe SCD with a history of at least two VOCs per year during the previous two years were eligible to participate in the study [104]. The safety and effectiveness of the trial were monitored for six months to two years after the infusion of CTX001. Preliminary results of the therapy demonstrated that the patients were free of VOCs, and clinically significant levels of HbF were found in the patients with no CTX001-related serious adverse events [103,105] (Table 1).

**Table 1.** Details of the pivotal clinical trials utilizing CRISPR/Cas9 gene therapy for β-hemoglobinopathies.

| Trial Description | Participants | Primary Endpoint | Baseline Characteristics for Patients with 3 Months Follow Up | Adverse Events | References |
|---|---|---|---|---|---|
| CLIMB THAL–111 Study Id No: CTX001-111 NCT No: NCT03655678 Title: A Phase 1/2/3 Study of the Safety and Efficacy of a Single Dose of Autologous CRISPR-Cas9 Modified CD34+ Human Hematopoietic Stem and Progenitor Cells (hHSPCs) in Subjects With Transfusion-Dependent β-Thalassemia. Pathology: TDT | Both male and female; Age limit: 12 years to 35 years; Number of estimated participants: 45; History of at least 100 mL/kg/year or ≥10 units/year of packed RBC transfusions in the prior 2 years; Homozygous β-thalassemia or compound heterozygous β-thalassemia including β-thalassemia/hemoglobin E (HbE) | Sustained transfusion reduction of 50% for ≥6 months, beginning 3 months after the CTX001 infusion | (n = 5) Median neutrophil engraftment occurred on day 32 after infusion of CTX001 Median platelet engraftment occurred on day 37 after infusion of CTX001 Total median Hb level: 11.5 g/dL Median HbF level: 8.4 g/dL | AE were reported in 1 patient with TDT: Headache, Haemophagocytic, Lymphohistiocytosis (HLH), Acute respiratory distress syndrome, Idiopathic pneumonia syndrome (All 4 of these AEs were resolved/clinically improved after 15 months of CTX001 infusion) | [64,106] |

**Table 1.** *Cont.*

| Trial Description | Participants | Primary Endpoint | Baseline Characteristics for Patients with 3 Months Follow Up | Adverse Events | References |
|---|---|---|---|---|---|
| CLIMB SCD—121 Study Id No: CTX001-121 NCT No: NCT03745287 Title: A Phase 1/2/3 Study to Evaluate the Safety and Efficacy of a Single Dose of Autologous CRISPR-Cas9 Modified CD34+ Human Hematopoietic Stem and Progenitor Cells (CTX001) in Subjects With Severe Sickle Cell Disease Pathology: SSCD | Both male and female; Age limit: 12 years to 35 years; Number of estimated participants: 45; Presence of previous indications of two or more severe vaso-occlusive episodes per year for a period of early two years; Occurrence of βS/βS or βS/β⁰ genotype | After 6 months from CTX001 infusion, ≥20% sustained level of HbF for ≥3 months No vaso-occlusive episodes during the 16.6 months after the infusion of CTX001 | (*n* = 2) Median neutrophil engraftment occurred on day 22 after infusion of CTX001 Median platelet engraftment occurred on day 30 after infusion of CTX001 Total median Hb level: 10.0 g/dL Median HbF level: 4.21 g/dL | 114 adverse events were identified in patient 2 with SCD, among which 3 were classified as serious adverse events. sepsis in the presence of neutropenia; Cholelithiasis; Abdominal pain All 3 adverse events were resolved upon treatment. Furthermore, intermittent, nonserious lymphopenia was observed | [64,107] |
| CLIMB 131 Study Id No: CTX001-131 NCT No: 04208529 Title: A Long-term Follow-up Study of Subjects With β-thalassemia or Sickle Cell Disease Treated With Autologous CRISPR-Cas9 Modified Hematopoietic Stem Cells (CTX001) Pathology: TDT | Both male and female; Age limit: 2 years and older (Child, Adult, Older Adult); Number of estimated participants: 114; Subjects must have received CTX001 infusion in a parent study (CTX001-111 or CTX001-121 or VX21-CTX001-141 or VX21-CTX001-151) | N/A | N/A | N/A | [108] |
| CLIMB 141 Study Id No: VX21-CTX001-141 NCT No: NCT05356195 Title: A Phase 3 Study to Evaluate the Safety and Efficacy of a Single Dose of CTX001 in Pediatric Subjects With Transfusion-Dependent β-Thalassemia Pathology: TDT | Both male and female; Age limit: 2 years to 11 years (Child); Number of estimated participants: 12; Homozygous or compound heterozygous β-thalassemia including β-thalassemia/hemoglobin E (HbE); History of at least 100 mL/kg/year of packed RBC transfusions in the prior 24 months | N/A | N/A | N/A | [109] |

**Table 1.** *Cont.*

| Trial Description | Participants | Primary Endpoint | Baseline Characteristics for Patients with 3 Months Follow Up | Adverse Events | References |
|---|---|---|---|---|---|
| CLIMB 151 Study Id No: VX21-CTX001-151 NCT No: NCT05329649 A Phase 3 Study to Evaluate the Safety and Efficacy of a Single Dose of CTX001 in Pediatric Subjects With Severe Sickle Cell Disease Pathology: SSCD | Both male and female; Age limit: 2 years to 11 years (Child); Number of estimated participants: 12; Presence of $\beta^S/\beta^S$ or $\beta^S/\beta^0$ genotype; Previous indications of two or more severe vaso-occlusive episodes per year for a period of early two years | N/A | N/A | N/A | [110] |

Abbreviations: TDT: transfusion-dependent β thalassemia; SSCD: severe sickle cell disease; N/A: not applicable; $\beta^S/\beta^S$: sickle cell disease genotype (HBB: c.20A > T).

The advancement of CTX001 therapy created a promising therapeutic approach for β-thalassemia. Previously, eligibility for CLIMB THAL-111 was limited to patients between ages 18–35 years but later changed to patients between ages 12–35 years with a defined history of at least 100 mL/kg of body weight/year of packed red blood cell transfusions during the previous two years [104]. CD34+ HSPCs were collected from patients through mobilization with filgrastim and plerixafor, followed by apheresis. These cells were then modified by CRISPR/Cas9 mediated genome editing. Here, a 20 nt long sgRNA was used to target the erythroid enhancer region of BCL11A while Cas9 performed the site-specific endonuclease activity, resulting in the successful disruption of this motif. The modified cells were then subjected to infusion after testing and quality control. Following the myeloablative conditioning with busulfan, patients received the CRISPR-modified CD34+ HSPCs as an intravenous one-time infusion (Figure 3). These patients were monitored for engraftment and hematopoietic recovery and adverse events along with the increment of HbF and total Hb levels [64,103,104] (Table 1).

CTX001 therapy has led to the disruption of the erythroid-specific enhancer binding site of BCL11A, resulting in the downregulation of BCL11A expression. As a result, transcriptional silencing of γ-globin by BCL11A is reversed, and the γ-globin expression is reactivated to increase HbF and total Hb levels in CTX001 received TDT patients. Maintenance of elevated levels of HbF and total Hb levels were also observed in patients over time [105,111]. During the 21.5 months after receiving CTX001 therapy, the very first patient showed 32 adverse events, but they were considered to be in the early stages of severity. Some patients in early clinical trials showed severe adverse events such as haemophagocytic lymphohistiocytosis and acute respiratory distress syndrome [64,103]. Overall, these adverse events were found to be correlated with busulfan myeloablation and autologous HSPC transplantation [104]. Other detrimental effects observed included pneumonia and sepsis with neutropenia, veno-occlusive disease and sinusoidal obstruction syndrome, abdominal discomfort, and cholelithiasis, as well as non-serious consequences of lymphopenia (Table 1) [64]. The occurrence of these effects could be attributed to the delay in the recuperation of lymphocytes due to CD4+ T cell enrichment after CTX001 therapy [64,112]. The CLIMB-Thal-111 study was formulated to appraise the safety and efficacy

of CTX001 dosing for patients of 12–35 years of age while endorsing long term-follow up protocols of up to 15 years (CLIMB-Thal-131) [113].

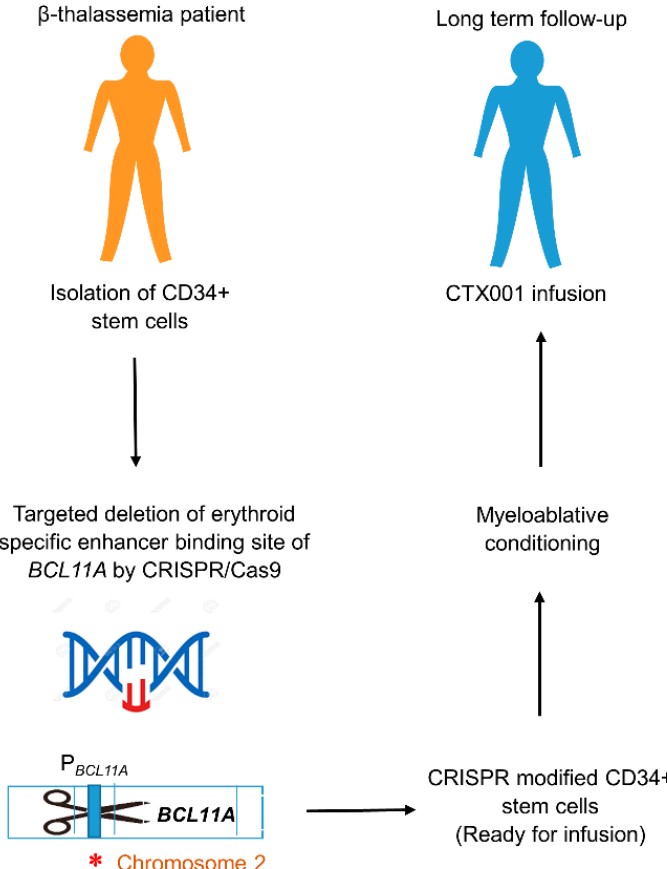

**Figure 3.** CTX001 therapy workflow. CTX001therapy is a potential one-time gene therapy for TDT and severe SCD that relies on CRISPR/Cas9 gene editing to facilitate the patient's own cells to produce high levels of HbF. CRISPR/Cas9 system targeting the deletion of erythroid-specific enhancer binding site of *BCL11A* is introduced into CD34+ stem cells from β-thalassemia patients, resulting in downregulation of *BCL11A* expression. Subsequent inactivation of BCL11A binding to the $P_{γ\text{-globin}}$ causes reactivation of the γ-globin gene, resulting in fetal globin switch. After quality controlling and clinical testing, CRISPR-modified CD34+ stem cells are introduced into β-thalassemia patient via autologous transplantation, along with myeloablative chemotherapy and immune-suppressive treatments [103–105]. $P_{BCL11A}$; Promoter region of *BCL11A* gene. ∗ (Asterisk symbol); Indel mutation introduced by CRISPR/Cas9.

In addition to CTX001 trials, ST-400 is another type of investigational ex vivo autologous cell therapy that utilizes gene editing to increase the production of HbF. It is a phase 1/2 clinical study that relies on Sangamo's Zinc Finger Nuclease (ZFN) technology, which facilitates targeted genome editing. ST-400 trials follow a similar strategy as CTX001 to disrupt the enhancer of the *BCL11A* gene, subsequently reactivating the γ-globin gene expression [114]. Nevertheless, ongoing CTX001 trials with positive outcomes suggest that this novel therapy has considerable potential to be used in treating TDT patients [44]. However, adverse events such as unintended off-target editing, inefficient gene delivery, and autoimmune reactions must receive considerable attention during clinical use. Hence, careful monitoring and follow-up of clinical efficacy, genotoxicity, and other safety protocols are imperative. Comprehensive pre-clinical studies and laboratory and computational-based techniques must be employed to identify the risk of observing possible adverse events [64]. Further progress in experimentation and understanding of the underlying

molecular concepts about the safety and efficacy of CTX001 therapy will shed light on therapy, improving the quality of life of patients.

## 5. Relative Merits

CRISPR-mediated site-specific gene editing technology has brought about a transformative and promising treatment method for TDT. This approach avoids the time-consuming steps of protein engineering, such as altering and generating site-specific nucleases that recognize a specific DNA sequence [115], while enhancing the target design simplicity and reducing implementation time. Therefore, CRISPR/Cas9 has proven to be a simple, efficient, and cost-effective gene therapy technique for inherited blood disorders [116]. Unlike viral-vector-mediated gene insertion, genome editing by CRISPR does not require the use of integrating vectors, since transitory production of a specific endonuclease is able to induce the required DNA cleavage.

Even though CRISPR technology alleviates some of the hindrances associated with traditional gene therapy, it has some drawbacks in the clinical setting. A major concern of this method is the higher frequency of unintended genome cleavages, resulting in off-target mutagenesis rendering unforeseen and undesirable outcomes during therapeutic trials. Off-target effects impose life-threatening risks due to deleterious genetic changes and deprivation of gene function, eventually leading to unfavorable phenotypes. At present, researchers are in the process of exploring different strategies to mitigate this issue [73,117]. Off-target effects were assessed using a variety of methodologies during CRISPR-gene therapy for β-thalassemia [118]. Several studies have shown that the use of high-fidelity enzymes and a double-nicking strategy can significantly decrease off-target activity while preserving acceptable efficiency for editing the target genome [118,119]. For instance, recently, an efficient Cas9 double nickase mediated gene targeting strategy was employed for iPSC of a patient with β-thalassemia mutation (IVSII-1 G > A) by integrating catalytic mutant Cas9D10A nickase with a pair of sgRNAs complementary to the target sequence [119,120]. The optimization of single guide RNAs can also improve the efficacy and accuracy of gene editing [117,121]. The application of in silico methods for predicting and quantifying off-targets is important in overcoming this problem [117,121,122].

However, appealing uses of this approach raise certain challenges on ethical, social, and safety protocols, specifically when utilizing it in the clinical platform [123]. The uncertainty of whether this therapy causes a permanent genetic alteration in the organism and whether the modified gene passes to subsequent generations is a major concern. Even if the genome is altered as planned and the intended functional problem is resolved at the specified time, it is uncertain whether the complicated interaction between genetic information and biological phenotypes would be clearly defined. Undesired clinical outcomes may arise due to uncertainty associated with genome modifications in complex biological systems creating dilemmas in ethical deliberations [123,124]. Therefore, comprehensive discussions on the societal ramifications involving scientists, legislators, and ethicists are required as the utilization and patenting of gene therapy techniques for treatment purposes has caused a rift within scientific communities [123]. However, it is noteworthy that each of these ongoing and prospective advancements in designing and enhancing CRISPR/Cas9-based systems for genome editing will undoubtedly revolutionize the field of medicine in the upcoming decade and will possibly provide a permanent cure for β-hemoglobinopathies.

## 6. Conclusions

CRISPR/Cas9 has paved the way to new therapeutic interventions for human genetic diseases, thus, proving to be a promising curative strategy for inherited blood disorders. Conventional therapies such as blood transfusion, iron chelation, and pharmaceutical drugs may temporarily ameliorate the clinical severity of β-thalassemia but do not provide a permanent curative option for this disease. At present, HSCT and viral-vector-based gene therapy are the only treatment options that offer a potential permanent cure for TDT. While the lack of donor histocompatibility reduces the chances of the use of HSCT as a

curative strategy, viruses used in gene therapies may increase cancer risk, limiting the use of these approaches as a curative treatment for TDT. Non-viral gene therapies such as liposomal gene delivery, polymeric gene delivery, and DNA microinjection seem attractive alternatives to viral vectors due to their safety advantages including reduced pathogenicity and easy low-cost production. Nonetheless, low transfection efficacy remains a critical bottleneck for the clinical application of non-viral gene delivery methods. Hence, the focus is now shifted towards CRISPR/Cas9 gene therapy as a promising therapeutic tool for β-hemoglobinopathies. CTX001 therapy that relies on ex vivo CRISPR/Cas9 gene editing was shown to improve the HbF level in CRISPR-edited autologous CD34+ HSPCs and may reduce the life-long blood transfusion requirement in TDT patients. Aside from legal and ethical considerations and the major challenge of off-target mutagenesis, ex vivo CRISPR/Cas9-mediated gene therapy offers great hope as a promising one-time curative therapeutic strategy for the treatment of β-thalassemia.

**Funding:** This research received no external funding.

**Acknowledgments:** The authors would like to express their heartfelt gratitude to N. V. Chandrasekharan, Department of Chemistry, University of Colombo for providing reading materials that greatly helped them to develop the idea of writing this manuscript. They would also express special thanks to Ramanee Wijesekara, Department of Chemistry, University of Colombo for providing insightful comments on the manuscript.

**Conflicts of Interest:** The authors declare no conflict of interest.

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
