# Peer review of "CRISPR Gene Therapy: A Promising One-Time Therapeutic Approach for Transfusion-Dependent β-Thalassemia—CRISPR-Cas9 Gene Editing for β-Thalassemia"

_thalassrep, doi:10.3390/thalassrep13010006_

Round 1

Reviewer 1 Report

Review manuscript from Udani et al describes an emerging CRISPR gene editing therapy for treatment of transfusion-dependent beta-thalassemia. This is indeed a promising approach in this exciting era of cell/gene therapies. Few more clinical trials and perhaps longitudinal studies are required to determine the future of this therapeutic strategy. 

The manuscript is well written and organized. Please find few minor revisions that need to be addressed prior publishing in Thalassemia reports.  

1. Please make sure the reference numbers are cited next to period or comma in the text. It is difficult to get if that is a number in the text or a reference number. 

2. In table 1, recruitment status, pathology, study type columns can be removed and be added as footnotes. So that the table becomes readable and also references in the table are not visible in the current form.

3. The text size is not uniform in manuscript. Please check. 

4. Suggest getting the permissions from original for the figures; ignore if those created by authors. 

5. Are there any other gene editing approaches that are currently in clinical trials?

6. Are these clinical trials are limited to any single continent population or being expanded throughout? Couple of lines in the text may need to be added with respect to treatment strategy and population subtypes. 

Author Response

Dear Reviewer,

Thank you very much for the review of our manuscript entitled:CRISPR gene therapy: A promising one-time therapeutic approach for transfusion-dependent β-thalassemia - CRISPR-Cas9 gene editing for β-thalassemia.

We sincerely appreciate all your valuable comments and suggestions, which helped us greatly improve the quality of the article. Our responses to your valuable comments are described in a point-to-point manner. Most of the required changes, suggested by you, have been introduced to the manuscript.

Comment

  1. In table 1, recruitment status, pathology, study type columns can be removed and be added as footnotes. So that the table becomes readable and also references in the table are not visible in the current form.

Response-We have removed some of the columns suggested by the reviewer to improve the clarity.

Comment

  1. The text size is not uniform in manuscript. Please check. 

Response-Attended.

Comment

  1. Are there any other gene editing approaches that are currently in clinical trials?

Response- We have briefly described the other gene editing approaches for TDT that are currently in clinical trials (e.g., ST-400). We did not add details about this therapy as the main focus of the article is CRISPR-edited gene therapies such as CTX001.

Comment 

  1. Are these clinical trials are limited to any single continent population or being expanded throughout? Couple of lines in the text may need to be added with respect to treatment strategy and population subtypes. 

Response-Attended

Reviewer 2 Report

This review shows the relevance of CRISPR-based gene therapy to treat transfusion-depended β-thalassemia (TDT), and the scientific steps that have been taken to achieve this goal. The authors first describe the classification of β‑thalassemia types based on HBB mutations. Standard treatment options and new therapeutic strategies, including gene therapy, for the management of β‑thalassemia are also discussed. Then, authors comprehensively describe the advances in CRISPR-based gene therapy to treat TDT and specifically the CTX001 therapy that relies on ex vivo CRISPR/Cas9 gene editing. Clinical trials results, advantages and disadvantages of CRISPR-based CTX001 therapy, ethical considerations and the challenge of off-target mutagenesis are also discussed. This is a very interesting review, well organized and detailed described.

Minor changes required

-Table1. The font size makes it difficult to read. Pherhaps, table 1 could be split in two.

-Line 221-222 (page 5). “CRISPR mediated adaptive immune systems were first discovered in bacteria that provide protection against invading foreign genetic elements 66–68”. A reference should be made to the paper by Mojica JF, in which he first described the CRISPR system.

- Lines 133, 166, 247, 286, 289, 330, 509. Extra spaces between words must be removed.

-Lines 353 to 389 (page 9). The font size must be changed.

-A list of abbreviations should be included.

Author Response

Dear Reviewer,

Thank you very much for the review of our manuscript entitled: CRISPR gene therapy: A promising one-time therapeutic approach for transfusion-dependent β-thalassemia - CRISPR-Cas9 gene editing for β-thalassemia.

We sincerely appreciate all your valuable comments and suggestions, which helped us greatly improve the quality of the article. Our responses to your valuable comments are described in a point-to-point manner. Most of the required changes, suggested by you, have been introduced to the manuscript.

-Table1. The font size makes it difficult to read. Pherhaps, table 1 could be split in two.

Response-We have removed some of the columns in the table to improve the clarity.

-Line 221-222 (page 5). “CRISPR mediated adaptive immune systems were first discovered in bacteria that provide protection against invading foreign genetic elements 66–68”. A reference should be made to the paper by Mojica JF, in which he first described the CRISPR system.

Response-Attended.

- Lines 133, 166, 247, 286, 289, 330, 509. Extra spaces between words must be removed.

Response-Attended.

-Lines 353 to 389 (page 9). The font size must be changed.

Response-Attended.

-A list of abbreviations should be included.

Response-We did not add a separate list of abbreviations as we described all the abbreviations in the text.

.
